# Whale Optimization Algorithm with a Hybrid Relation Vector Machine: A Highly Robust Respiratory Rate Prediction Model Using Photoplethysmography Signals

**DOI:** 10.3390/diagnostics13050913

**Published:** 2023-02-28

**Authors:** Xuhao Dong, Ziyi Wang, Liangli Cao, Zhencheng Chen, Yongbo Liang

**Affiliations:** 1School of Life and Environmental Sciences, Guilin University of Electronic Technology, Guilin 541004, China; 2Guangxi Colleges and Universities Key Laboratory of Biomedical Sensors and Intelligent Instruments, Guilin 541004, China; 3Guangxi Engineering Technology Research Center of Human Physiological Information Noninvasive Detection, Guilin 541004, China

**Keywords:** respiratory rate, photoplethysmography signal, hybrid relation vector machine (HRVM), whale optimization algorithm (WOA), ensemble empirical mode decomposition with principal component analysis (EEMD-PCA)

## Abstract

Due to the simplicity and convenience of PPG signal acquisition, the detection of the respiration rate based on the PPG signal is more suitable for dynamic monitoring than the impedance spirometry method, but it is challenging to achieve accurate predictions from low-signal-quality PPG signals, especially in intensive-care patients with weak PPG signals. The goal of this study was to construct a simple model for respiration rate estimation based on PPG signals using a machine-learning approach fusing signal quality metrics to improve the accuracy of estimation despite the low-signal-quality PPG signals. In this study, we propose a method based on the whale optimization algorithm (WOA) with a hybrid relation vector machine (HRVM) to construct a highly robust model considering signal quality factors to estimate RR from PPG signals in real time. To detect the performance of the proposed model, we simultaneously recorded PPG signals and impedance respiratory rates obtained from the BIDMC dataset. The results of the respiration rate prediction model proposed in this study showed that the MAE and RMSE were 0.71 and 0.99 breaths/min, respectively, in the training set, and 1.24 and 1.79 breaths/min, respectively, in the test set. Compared without taking signal quality factors into account, MAE and RMSE are reduced by 1.28 and 1.67 breaths/min, respectively, in the training set, and reduced by 0.62 and 0.65 breaths/min in the test set. Even in the nonnormal breathing range below 12 bpm and above 24 bpm, the MAE reached 2.68 and 4.28 breaths/min, respectively, and the RMSE reached 3.52 and 5.01 breaths/min, respectively. The results show that the model that considers the PPG signal quality and respiratory quality proposed in this study has obvious advantages and application potential in predicting the respiration rate to cope with the problem of low signal quality.

## 1. Introduction

Impedance spirometry is a clinically important method for measuring the respiratory rates of patients; however, it is not very convenient and comfortable for patients and is especially not suitable for dynamic monitoring [1]. To overcome these restrictions, studies have given more attention to physiological signals, such as electrocardiogram (ECG) and photoplethysmography (PPG) signals. However, PPG signals are more attractive than ECG signals because of their simplicity, portability, and low number of sensors [2,3].

Due to the low cost and portability of PPG signal acquisition, the continuous, non-invasive, and reliable monitoring of patients’ respiratory rate based on PPG signals has attracted many researchers in recent years [4], which will help monitor primary health status and contribute to the diagnosis of cardiorespiratory diseases, including pneumonia [5] and obstructive sleep apnoea (OSA) [6], and physiological situations, such as hypercarbia [7] and pulmonary embolism [8]. However, for critically ill patients in hospital care situations, where human respiration is relatively weaker and venous blood is at a very low pressure, flowing with the frequency of movement [9], the pulsatile alternating current (AC) component is also easily affected by physiological activity and movement, causing the PPG signals to contain more noise than physiological information [10], which would make estimating the respiratory rate based on PPG signals specifically challenging.

In the 1990s, the method of estimating the respiration rate (RR) based on PPG signals was investigated by Nakajima et al. [11], who developed a digital filter technology to estimate the RR from PPG signals. This method proved the feasibility of RR estimation based on PPG, but lacked universality. To solve this problem, Karlen et al. [12] used an intelligent fusion algorithm to fuse three respiration-modulated changes, including pulse amplitude modulation (AM), frequency modulation (FM), and baseline wander (BW), from PPG, which estimated a relatively accurate RR. Similarly, Meredith et al. [13] explained that respiratory components are reflected by the AM, FM, and BW of PPG signals. To reduce the effect of poor respiratory modulation on the accuracy of the final estimated respiration rate, Birrenkott et al. [14] proposed three respiratory quality indices (RQIs), which set an adjustable threshold to fuse the respiratory rate estimated by three respiratory modulations, on the dataset of elective surgery or routine anaesthesia (CapnoBase Dataset), which showed good results with a mean absolute error (MAE) of only 0.71 ± 0.89 bpm. However, for critically ill patients during hospital care (BIDMC dataset), the MAE reached 3.12 ± 4.39 bpm. Selvakumar et al. [3] also showed that the BIDMC dataset was more challenging.

Faint physiological conditions and the more complex pulse wave morphology in critically ill patients lead to the inaccurate extraction of respiratory modulation [15]. To enhance the robustness of the respiration rate estimation algorithm. Ambekar el al. [16] used a data-driven algorithm, ensemble empirical mode decomposition (EEMD), to obtain the RR from PPG signals. Compared to the empirical mode decomposition algorithm (EMD), EEMD overcomes the problem of modal aliasing [17]. Adami et al. [18] and Mohammod et al. [19] compared the performance of EMD and its derivative algorithms, such as EEMD, CEEMD, CEEMDAN, and ICEEMDAN, to decompose the respiratory component from PPG signals and demonstrated that both EEMD and CEEMDAN have better performance but that the EEMD method has a lower computational cost than CEEMDAN. To further isolate the respiratory components from the selected inherent modal functions (IMF), a Kalman filter (KF) was used to exclude other components [20]. Considering the nonlinearity of the PPG signal, Adami et al. [18] introduced the PPG signal quality indicator as a way to adjust the Kalman gain to implement extended Kalman filtering (EKF). However, these methods are dependent on the stability of the previous window, and if the respiratory rate error in the first window is large, it will significantly increase the overall estimation error level. The principal component analysis (PCA) method avoids this problem and does not require parameter adjustment [21]. Mohammod et al. proposed an EEMD-PCA method, demonstrated its advantages for respiration rate estimation based on PPG signals [22], and explained the failure as high-amplitude noise in the RR band range [23]. Machine-learning methods can be used to correct respiration rate estimation errors due to these high-amplitude noises [24], For example, the RRWaveNet [25] and CAGAB [26] methods demonstrate the feasibility of using machine-learning methods to improve the accuracy of PPG-based respiratory rate detection.

In summary, both PPG signal quality and respiratory signal quality influence the accuracy of the PPG-based estimation of respiration rate to further reduce their effect on respiration rate estimation. In this paper, we use machine-learning methods to construct a respiration rate estimation model by fusing PPG signal quality indices and respiratory quality indices (RQIs) to limit the influences of other physiological activities and noise on respiration rate estimation based on PPG, developing a respiration rate estimation model with high accuracy and robustness.

The remainder of this study is summarized as follows: Section 2 introduces the dataset and respiration rate prediction model construction process with model performance evaluation metrics. In Section 3, the performance of the model proposed in this study on the BIDMC dataset is reported, followed by a discussion of the advantages and disadvantages of the model in Section 4. Section 5 concludes the study and provides recommendations for future work.

## 2. Materials and Methods

Figure 1 shows a clear block diagram of the respiration rate model construction process. The whole process is divided into three stages: (a) pre-extracted respiratory wave and respiratory rate, (b) signal quality index calculation, and (c) respiratory rate prediction model construction.

In the first stage, the EEMD-PCA method was selected to pre-extract the respiration wave and respiration rate. In the second stage, the PPG signal quality indices and RQIs were calculated for further processing. The HRVM method was employed to fuse the respiration rate estimated by the EEMD-PCA method (RR_P_) and the signal quality indicators that have an impact on the estimation of RR_P_ for RR prediction model construction, and the kernel parameters were optimized by the WOA algorithm to prevent falling into a local optimum. To evaluate the performance of the constructed model, it is tested with the BIDMC dataset.

### 2.1. Database

The BIDMC dataset was collected from 53 subjects (20 males and 33 females; age range: 19–90 years old) and acquired from critically ill patients during hospital care at the Beth Israel Deaconess Medical Centre (Boston, MA, USA). For each subject, over an 8 min duration, each subject contains physiological signals that are sampled at 125 Hz, such as PPG signals, impedance respiratory signals, and electrocardiogram (ECG) signals; simultaneously, reference physiological parameters such as respiratory rate (RR) and heart rate (HR) are sampled at 1 Hz. Two annotators manually annotated the start and end time points of each single respiration in all recordings via the impedance respiration signal, and the corresponding PPG signal segment with a 1-s difference in the duration of single respiration annotated by the two annotators was removed. Due to a severe loss of the reference respiration rate in the 13th subject, the remaining 52 subjects were retained with the PPG signal split into 8-s nonoverlapping windows with a 32-s length. This process resulted in 2719 (93.4%) windows being retained. The distribution of the impedance respiration rate (RR_I_) values for all windows is shown in Figure 2, which reveals that the distribution of the reference respiration rate ranges from 3 to 30 bpm, mainly between 16 and 20 bpm, follows a regular distribution, and reflects well the real-world respiration rate distribution. In this study, the dataset was randomly divided into a training set with a validation part (70%) and a test set (30%).

### 2.2. Pre-Extracted Respiratory Wave and Respiratory Rate by the EEMD-PCA Method

EEMD-PCA is a novel, data-driven method for estimating the respiration rate based on PPG signals that was proposed by Mohammod et al. [22]. In this study, the pre-extracted respiratory waves and RR_P_ will be extracted by this method for use in the next stage. The pre-extraction process is subdivided into four steps: (a) EEMD is applied to PPG signals to separate the respiratory components and other components, (b) intrinsic mode functions (IMFs) dominated by respiratory components are selected for further processing, (c) the selected IMFs are used to reconstruct the respiratory waves and are further denoised with principal component analysis (PCA), and (d) fast Fourier transform (FFT) is applied to the pre-extracted respiratory waves from the previous step to calculate the RR_P_. Figure 3 shows the time domain (Figure 3a) and frequency domain (Figure 3b) of the impedance respiratory signals and the pre-extracted respiratory waves for the first 32-s window of subject BIDMC 10. They have strong consistency in the waveform period in the time domain, and the main frequency components are similar in the frequency domain.

FFT was applied to the pre-extracted respiratory signals, which are dominated by respiration, and the frequency corresponding to the maximum peak of the spectrogram is expressed as the frequency corresponding to the respiration rate and then converted to the $RR_{P}$ using Formula (1).
(1)$$RR_{P}=f_{RR_{P}}*60\left(breaths/min\right)$$

### 2.3. Signal Quality Index Calculation

Both PPG signal quality indices [18,20,25] and respiration quality indices (RQIs) [14,26] affect the accuracy of RR estimation based on PPG signals. An optimal PPG signal quality index ($S_{SQI}$) and three typical RQIs ($Q_{R1}$, $Q_{R2}$, and $Q_{R3}$) are calculated in this section to fuse the RR_P_ to reduce the error of RR estimation and enhance the robustness of the algorithm proposed in this paper.

#### 2.3.1. PPG Signal Quality Index (SQI) Calculation

Skewness is a measure of the symmetry of the probability distribution. Mohamed et al. [27] discovered that the skewness value of a 2-s PPG signal significantly varies with the change in the quality of the PPG signals, with an accuracy of 82.86% in determining between high-quality PPG signals and damaged unusable PPG signals, which is calculated by Formula (2).
(2)$$S_{SQI}=\frac{1}{N}\sum_{i=1}^{N}{\left(x_{i}-{\hat{\mu }}_{x}/\sigma \right)}^{3}$$
where $x_{i}$ is the $i^{th}$ sample point value of PPG, ${\hat{\mu }}_{x}$ and σ are the empirical estimates of the mean and standard deviation of $x_{i}$, respectively, and N is the number of samples in the PPG signals.

For each 32-s PPG signal with a two-second nonoverlapping sliding window, a total of 16 skewness values are calculated, and the average of 16 $S_{SQI}$ represents the overall quality level of the 32-s PPG signal for that segment. The specific process is expressed as follows:(3)$$\bar{S_{SQI}}=\frac{1}{n}\sum_{w=1}^{n}S_{SQI}\left[w\right]$$
where $S_{SQI}\left[w\right]$ denotes the $S_{SQI}$ of the PPG signals for the $w^{th}$ 2-s window and n is the number of windows. $\bar{S_{SQI}}$ denotes the quality level of the PPG signal for each 32-s window.

#### 2.3.2. Respiratory Quality Index (RQI) Calculation

The autocorrelation RQI, FFT RQI, and autoregression RQI ($Q_{R1}$, $Q_{R2}$, and $Q_{R3}$) were proposed by Birrenkott et al. [14], who directly calculated their RQIs on the PPG signal after filter processing with a fixed cut-off frequency and down-sampled to 4 Hz, which still contains much low-frequency motion noise and cardiac components. In contrast, this paper will calculate the three RQIs on the pre-extracted respiration waves down-sampled to 4 Hz, which are dominated by respiratory components that are more reflective of respiratory signal quality.

### 2.4. WOA-HRVM Model

In the previous stage, the RR_P_, $\bar{S_{SQI}}$, and three RQIs were obtained and used as features in this stage, and the RR_I_ corresponding to each window was applied as labels. The WOA-HRVM [28] method was applied to the training set to construct an RR prediction model, and the testing set was employed to evaluate the performance of the model.

As a highly sparse model that provides probabilistic predictions by Bayesian inference, the central idea of related vector machines (RVMs) is to obtain the correlation vectors and weights by maximizing the marginal likelihood [29]. RVMs are often utilized as a machine-learning method for regression prediction, and its kernel function and kernel parameters are adjusted according to the requirements, which are also important parameters affecting the final regression performance. To improve the performance of the regression, a hybrid relation vector machine (HRVM) was employed in this paper. Since convex combinations of finitely many elementary kernel functions can always generate optimal kernels, hybrid kernel learning methods are more efficient than single kernel learning methods [30]. The multiple heterogeneous kernel learning method is defined as
(4)$$K\left(x_{i},x\right)=\sum_{m=1}^{M}d_{m}K_{m}\left(x_{i},x\right)$$
where $d_{m}$ is the weight of the $m^{th}$ kernel function with $d_{m}$ ≥ 0, and $K\left(x_{i},x\right)$ denotes the $m^{th}$ kernel function, Gaussian kernel function, sigmoid kernel function, polynomial kernel function, and Laplacian kernel function as common kernel functions used in this study.

In addition, the initial values of the kernel parameters are highly random; the convergence of the regression model constructed based on HRVM will be greatly affected as a result, and it is easy to fall into the local optimum. Aimed at the limitations of the HRVM algorithm, the whale optimization algorithm (WOA) [31] has the advantages of few adjustment parameters, simple operation, and strong ability for a global search. The optimal parameters and weights of the kernel function are obtained by continuous iteration of the WOA algorithm to prevent local optimality, so the respiration rate model proposed in this study based on the WOA-HRVM algorithm can be represented by Formula (4), it is a hybrid function consisting of Gaussian kernel function, sigmoid kernel function, polynomial kernel function and Laplace kernel function.

### 2.5. Performance Measurement

The ability of our RR prediction model was assessed using three methods: (i) Bland–Altman plot: the plot visualizes the consistency of the predicted respiration rate by the model proposed in this study (RR_M_) with the RR_I_; (ii) mean absolute error (MAE): the accuracy of the model was demonstrated by averaging the absolute value of the difference between RR_M_ and RR_I_ over all windows; and (iii) root-mean-square error (RMSE): RMSE is used to reflect the precision of the model proposed in this study; it is very sensitive to the very large or very small errors of the RR_M_ compared to RR_I_.

## 3. Results

The model in this study was constructed and tested based on MATLAB 2020a (MathWorks, Natick, MA, USA). The Bland–Altman plot visualizes the relationship among RR_P_ (RR estimated by the EEMD-PCA method), RR_M_ (RR estimated by the prediction model proposed in this study), and RR_I_ in Figure 4. In the training set, the difference in RR_P_ and RR_I_ was 0.07 bpm, with limits of agreement from −5.138 to 5.278 bpm, and the difference in RR_M_ and RR_I_ was almost 0 bpm, with smaller limits of agreement from −1.930 to 1.930 bpm. In the test set, the difference in RR_P_ and RR_I_ was 0.121 bpm, with limits of agreement from −4.660 to 4.906 bpm. The difference in RR_M_ and RR_I_ is only −0.015 bpm, with narrowed limits of agreement from −3.564 to 3.533 bpm.

The figure includes a total of three lines for each method, with the middle line indicating the mean of the differences, and the upper and lower lines showing the upper and lower limits, respectively, of the 95% consistency limits (mean ± 1.96SD). The closer the line showing the mean of the differences is to 0 bpm, the higher the agreement between the two measurement methods and the smaller the 95% confidence interval. The closer the method is to the impedance respiration test, the higher the clinical acceptability. Therefore, Figure 4 and Table 1 show that, compared to the EEMD-PCA method, the respiration rate estimated by the proposed method is more consistent with the respiration rate measured by the impedance spirometry.

Table 2 shows the MAE and RMSE of the PPG-derived RR (RR_P_ and RR_M_) with RR_I_ in the training set and test set. Even in the test set, the MAE and RMSE of the respiration rate prediction model proposed in this study are only 1.24 and 1.79 bpm, respectively, which are 0.62 lower and 0.65 bpm lower, respectively, than those of the EEMD-PCA method.

Figure 5 illustrates the performance of the proposed method for the continuous monitoring of RR based on PPG signal segments at different reference respiration rates in different people. The top half shows the training set results, and the bottom half shows the test set results, both for an 8 min duration.

The PPG signal segments from different people are mixed and have many sudden changes in RR, so we can check the capability of the respiration rate prediction model proposed in this study in tracking sudden changes and adaptability among different people. As shown in Figure 5, the proposed method shows good performance for the continuous detection of the respiration rate with mixed PPG signals at different respiration rates in different people. The proposed method is capable of estimating the sharp change in RR better than the EEMD-PCA algorithm.

Considering the performance in different respiratory rate ranges, Table 3 shows the performance in different respiratory rate ranges in the training set and testing set of the proposed model. According to Table 3, both the training set and the test set show good performance in the normal respiratory rate ranges of 12–16 bpm, 17–20 bpm, and 21–24 bpm. Even in the test set, the MAE is less than 2 bpm for the respiration rate prediction model proposed in this study, especially in the ranges of 17–20 bpm and 21–24 bpm, and the MAE decreases nearly twofold. In the range of human respiratory rates that are too fast or too slow (<12 bpm and >24 bpm), the MAE on the test set reaches 2.68 bpm and 4.28 bpm, respectively. However, compared to the EEMD-PCA method, the MAE still decreased by 3.57 bpm and 2.34 bpm.

## 4. Discussion

In this study, we consider both the PPG signal quality and respiration signal quality to estimate the respiration rate based on PPG signals and validate its accuracy and robustness on the BIDMC dataset. This proposed model is developed based on the HRVM and WOA algorithm. The use of hybrid kernel functions allows an exploration of the relationship between the error in the respiration rate estimated by the EEMD-PCA method and the signal quality indicators in a wider range of dimensions, and the WOA algorithm avoids falling into the local optimum by continuously iterating to identify the most suitable kernel function width and weight parameters. The respiration rate prediction model proposed in this study has the advantages of both a local kernel function and nonlocal kernel function. Therefore, the method has a higher accuracy in RR detection compared with other methods.

Previous studies used fixed threshold filters to pre-process PPG signals, which will inevitably filter out some respiration information. Therefore, we did not use any filters to pre-process the PPG signal. Instead, the data-driven EEMD-PCA method was directly utilized to exclude motion and cardiac noise and to pre-extract respiratory waves. In addition, the EEMD method is robust to noise, and PCA further reduces noise and cardiovascular signal interference based on variance, making the initially extracted respiratory signal highly reliable. Therefore, RQIs are calculated based on the pre-extracted respiratory waves better than the pre-processed PPG signal.

The complexity of the physiological condition of critically ill patients and the uncertainty of external noise produce complex changes in the error rate of the RR estimated by the EEMD-PCA method. Due to the differences in the human body, for different subjects, there is a significantly different Pearson correlation coefficient between the error rate of the RR estimated by the EEMD-PCA method and the four signal quality indices. Table 4 presents the R1, R2, R3, and R4 for 52 subjects, from which we observe that the R1, R2, R3 and R4 are significantly different for different subjects. For example, the correlation of four signal quality indicators of Subject 01 with the error in the respiration rate estimated based on the EEMD-PCA method is clearly more relevant than that of Subject 02. In addition, the sensitivity of different signal quality indicators differed for the same subject; for example, for Subject 04, compared to other signal quality indices, $Q_{R2}$ was not as relevant, while for Subject 01, it was $Q_{R1}$ that was less relevant. However, the mean values of these four correlation coefficients show that none of these four signal quality indicators is better than the other three signal quality indicators. To reduce the influence of signal quality indicators with a low correlation with the respiration rate estimation, the current state-of-the-art approach is to improve the accuracy of the estimates at the expense of discarding unusable data by ‘intelligent fusion’ methods. RQI Fusion reduces the percentage of discards by setting an adjustable signal quality indicator threshold. In this paper, we use the sparsity of the HRVM algorithm to select a certain percentage of data from the training set for the model construction and optimization by the WOA algorithm. This method is data-driven to determine the percentage of discards without setting a threshold parameter, and it is only necessary to give an objective function that yields an estimated respiration rate that is closest to the impedance respiration rate, which provides better robustness than other RR estimation methods.

In this paper, an end-to-end respiration rate prediction model is constructed. The advantages of the model proposed in this paper, in comparison with the end-to-end respiration rate prediction methods based on PPG signals proposed by other researchers, are shown in Table 5. According to our results and those of other authors on the BIDMC dataset in recent years, the MAE and RMSE of the model proposed in this paper on the test set are only 1.24 bpm and 1.79 bpm, respectively, which are much lower than those of other methods. As shown in Table 6, although the framework proposed in [18] and the EEMD + KF method [20] both show better results, the framework is too complicated to calculate; each time, it needs to use the EMD and DWT methods to calculate seven different predicted respiratory waves to fuse, and it takes 30 s to update the RR, 22 s more than the model proposed in this paper. The reason for choosing 8 s to update the breathing rate in this paper is based on matlab2020a with the EEMD-PCA method to decompose a 32-s PPG signal to extract the predicted respiratory rate and respiratory waves, plus the time to calculate four signal quality indicators is close to 8 s. Although the method in [20] is simpler to calculate, it relies on the signal quality of the first PPG window, and the error in estimating the respiration rate in the first window can lead to large errors in all subsequent windows. Neither the conventional respiratory-modulation-based methods in [14,32] in the paper showed good results, which is caused by the challenging respiratory modulation extraction when the signal quality is poor. For some advanced machine-learning methods or deep learning methods [26,27,32,33], they are affected by the accuracy of feature extraction or feature selection, resulting in their outcomes not being better than this paper’s. This paper uses the EEMD-PCA method to pre-extract the respiration rate and respiration wave, which not only avoids the challenging and inaccurate extraction based on the traditional respiration modulation, but also improves the accuracy and robustness of the respiration rate prediction by incorporating signal quality factors into the respiration prediction model using machine-learning methods. The RMSE of the CAGBA method was much lower than that of the other methods as only 20 subjects were selected.

To balance the continuity of the respiration rate detection and the accuracy of the respiration rate estimation, the appropriate PPG signal length is also important. It is evident from recent literature that the performance of respiration rate detection algorithms decreases as the PPG signal data length decreases. It is well-known that short data lengths are important for real-time respiratory rate detection in critical care or wearable devices, but a PPG signal that is too short is not conducive to accurate respiratory rate detection. In [34], the authors concluded that a length of 32 s is the most stable and shortest length for extracting respiratory signals based on PPG signals. Table 6 compared with other respiration rate estimation methods in recent years, the model proposed in this study showed better robustness and accuracy in estimating RR than other existing methods. A limitation of this study is the method for calculating PPG signal quality indicators and respiratory signal quality indicators. We calculated four signal quality indicators and tested them on different people. For some subjects, the sensitivity was poor, and a more sensitive signal quality index should be investigated. The choice of kernel function and the number of iterations of the WOA algorithm are also key factors affecting the accuracy of the final respiration rate prediction model. Other kinds of kernel functions and larger numbers of iterations should continue to be explored. In addition, the model was only tested on the BIDMC dataset; other datasets or an autonomous collection of real-world data should be collected by the latter to further validate the stability of the proposed method. The advantage of the proposed method is the end-to-end estimation of the respiration rate based only on PPG signals without a complex parameter adjustment, and the performance is significantly improved compared to other respiration rate estimation methods.

**Table 6 diagnostics-13-00913-t006:** Performance comparison of different PPG-signal-based respiration rate detection methods on BIDMC dataset.

References	Method	Length (sec)	Subjects	Overlap (sec)	MAE	RMSE	Dis (%)
This study	WOA-HRVM	32	53	24	1.24	1.79	6.6%
Adami [18]	EMD and DWT + EKF	60	53	30	0.73	-	-
Pongpanut [25]	RRWaveNet	32	53	-	1.62	-	1.9%
Sharma [20]	EEMD + KF	32	53	29	1.90	-	-
Aqajari [33]	CycleGAN	30	53	-	1.90	-	-
Lee [26]	CAGBA	32	20	0	1.94	0.61	62.26%
Bian [32]	Deep learning	60	53	59	2.50	-	-
Karlen [12]	SmartQualityFusion method	60	53	-	2.60	-	-
Birrenkott [14]	RQI calculation and fusion	32	53	17	3.12	4.39	23.2%

Notes: Dis (%) indicates the percentage of discarded data in the entire dataset.

## 5. Conclusions

In this paper, we used the WOA-HRVM method to fuse the PPG signal quality and respiratory signal quality with the respiratory rate estimated based on the EEMD-PCA method to construct a highly accurate and robust respiratory rate prediction model based on the PPG signal. The method is data-driven and does not require complex parameter tuning, which affects the stability of the respiratory rate prediction model, and overcomes the problem of difficult and inaccurate extraction when the signal quality is poor with traditional respiratory modulation methods. It also does not require the extraction of PPG morphological features and screening with feature selection methods as with other machine-learning methods or deep learning methods, and the final performance is affected by the accuracy of feature recognition and the performance of feature selection methods. After comparing the performance of the PPG-signal-based estimation of the respiration rate on the BIDMC dataset with that of previous investigators, the proposed methods showed more accurate results in estimating the RR than other existing methods for subjects from the BIDMC dataset with a short data length and a 32-s PPG signal. In future studies, we will validate the method using other datasets or an autonomous collection of real-world data in large cohorts of short data lengths while exploring more effective PPG and respiratory signal quality metrics to further improve the accuracy of the respiratory rate prediction model so that it can eventually be applied to the real-time detection of the respiratory rate on wearable devices or be utilized instead of impedance detection for the real-time detection of the respiratory rate in patients under intensive care. It promotes the development of portability for the real-time respiratory detection of patients under intensive care, and has a very important theoretical value for realizing the real-time detection of the respiratory rate in wearable devices and telemedicine, and improving the accuracy of respiratory rate measurement.

## Figures and Tables

**Figure 1 diagnostics-13-00913-f001:**
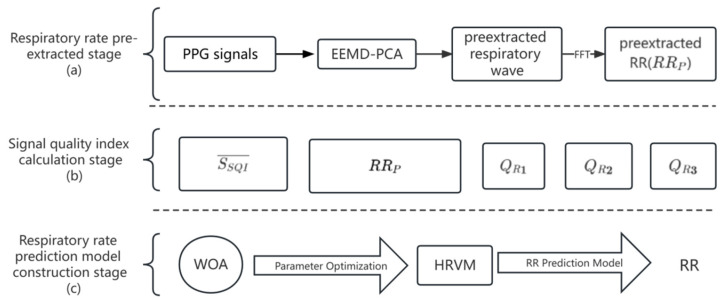
Flow chart of respiration rate prediction model construction.

**Figure 2 diagnostics-13-00913-f002:**
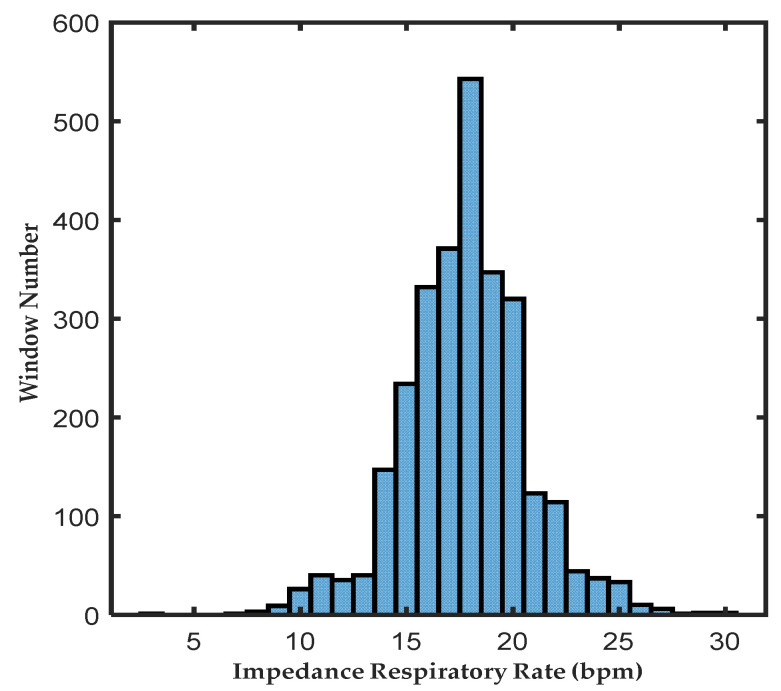
Distribution of each 32-s PPG window corresponding to the RR_I_. The horizontal axis indicates the value of the RR_I_ for each 32-s window, and the vertical axis indicates the number of windows at the corresponding RR_I_.

**Figure 3 diagnostics-13-00913-f003:**
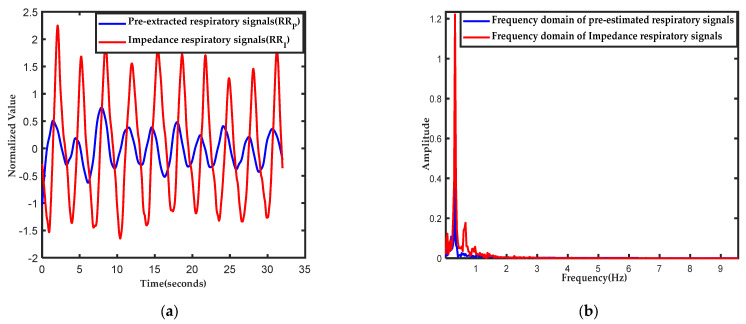
(**a**) Time domain comparison of impedance respiratory signals with pre-extracted respiratory signals based on the EEMD-PCA method. (**b**) Frequency domain comparison of impedance respiratory signals with pre-extracted respiratory signals based on the EEMD-PCA method. The blue line represents the pre-extracted respiratory signals in the time-frequency domain, and the red line represents the impedance respiratory signals in the time-frequency domain.

**Figure 4 diagnostics-13-00913-f004:**
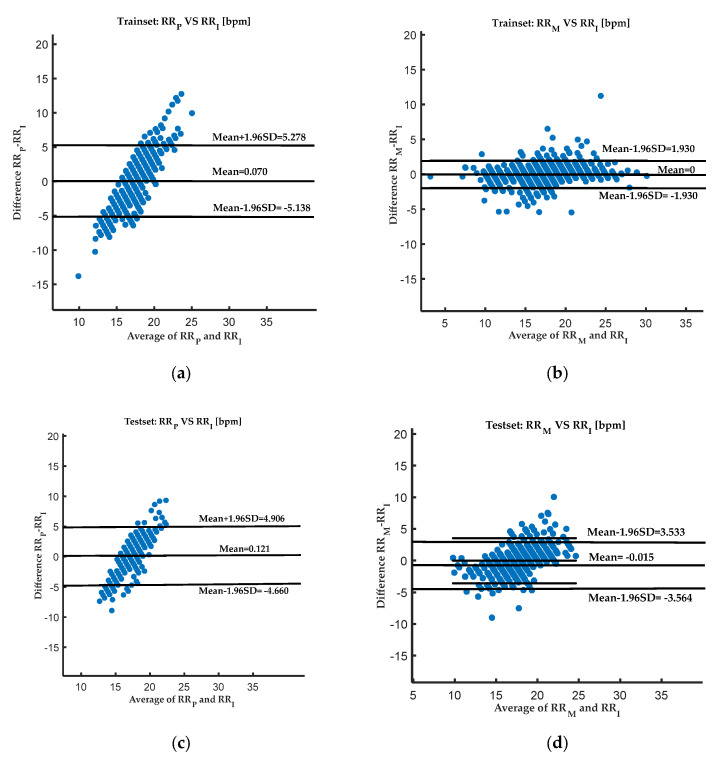
Bland–Altman plot of the RR_M_ (RR estimated by the prediction model proposed in this study), RR_P_ (RR estimated by the EEMD-PCA method), and RR_I_ for the training set and test set. The figure includes a total of 3 lines for each method, with the middle line indicating the mean of the differences, and the upper and lower lines showing the upper and lower limits, respectively, of the 95% consistency limits. The horizontal axis represents the average of RR_P_ and RR_I_ or the average of RR_M_ and RR_I_, and the vertical axis represents the difference, RR_P_-RR_I_ or RR_M_-RR_I_: (**a**) Bland–Altman plot of the RR_M_ and RR_I_ in the training set, (**b**) Bland–Altman plot of the RR_p_ and RR_I_ in the training set, (**c**) Bland–Altman plot of the RR_M_ and RR_I_ in the test set, and (**d**) Bland–Altman plot of the RR_p_ and RR_I_ in the test set.

**Figure 5 diagnostics-13-00913-f005:**
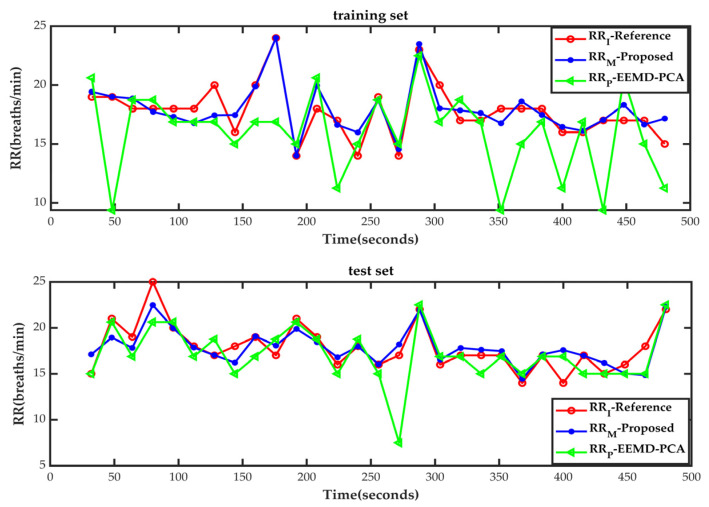
The EEMD-PCA method and respiration rate prediction model proposed in this study with different subjects and time windows for the estimation of mutant respiration rates were assessed for consistency with impedance respiration rates. (The red dotted line represents the referenced impedance respiration rate (RR_I_), the blue dotted line denotes the respiration rate predicted by the respiration rate prediction model proposed in this study (RR_M_), and the green dotted line indicates the respiration rate estimated by the EEMD-PCA method (RR_P_)).

**Table 1 diagnostics-13-00913-t001:** A literature review of respiratory rate detection based on PPG.

References	Database	Subjects	Methodology	Innovation	Drawbacks
Nakajima [11]	Self-collection	11	Digital filtering technique	Real-time PPG-based respiration rate detection	Lack of universal applicability
Karlen [12]	CapnoBase	94	Smart fusion method	Set discard thresholds with PPG signal quality metrics	Discard 45% of data
Birrenkott [14]	CapnoBase	42	RQI calculation and fusion	Adjustable threshold value to change accuracy	Inaccurate estimation of low-quality PPG signals
Selvakumar [3]	CapnoBase	42	RIAV based on FFT	Respiration rate detection on low-cost hardware	Low accuracy in detecting too-fast breathing
Sharma [20]	BIDMC	53	EEMD + KF	Kalman filtering is applied to the reconstructed signal	KF is not suitable for non-linear PPG signals
Adami [18]	BIDMC	53	CEEMDAN + DWT + EKF	Leverage time and frequency domain information	Framework calculation is too complicated
Mohammad [19]	MIMIC	121	EMD family and PCA	Free from parameter selection	Sensitivity to high-amplitude noise in the respiratory range
Shuzan [24]	VORTAL	39	Machine-learning model	Hyperparameter optimization	Tested only on resting young people
Pongpanut [25]	BIDMC	53	RRWaveNet	Improve model robustness using transfer learning	Discarded low-quality signals by SQI metric

**Table 2 diagnostics-13-00913-t002:** Mean of the differences, and the upper and lower limits of the 95% consistency limits for the RR_M_ and RR_P_ with the RR_I_ in the training set and test set.

Dataset	Method	Mean	Mean +1.96SD	Mean −1.96SD
training set	This study	0	1.930	−1.930
	EEMD-PCA	0.070	5.278	−5.138
test set	This study	−0.015	3.533	−3.564
	EEMD-PCA	0.121	4.906	−4.660

**Table 3 diagnostics-13-00913-t003:** Mean absolute error (MAE) and root mean square error (RMSE) for the RR_M_ (RR estimated by the prediction model proposed in this study) and RR_P_ (RR estimated by the EEMD-PCA method) with the RR_I_ in the training set and testing set.

Dataset	Method	MAE	RMSE
training set	This study	0.71	0.99
	EEMD-PCA	1.99	2.66
test set	This study	1.24	1.79
	EEMD-PCA	1.86	2.44

**Table 4 diagnostics-13-00913-t004:** Prediction performance in different ranges of respiratory rates.

RR_I_ (bpm)	Training Set			Test Set		
	N	MAE_this study_ [MAE_EEMD-PCA_]	RMSE_this study_ [RMSE_EEMD-PCA_]	N	MAE_this study_ [MAE_EEMD-PCA_]	RMSE_this study_ [RMSE_EEMD-PCA_]
below 12	61	1.08 [6.40]	1.56 [6.57]	16	2.68 [6.25]	3.52 [6.34]
12–16	563	0.79 [2.34]	1.08 [2.65]	202	1.45 [2.21]	1.90 [2.50]
17–20	1118	0.63 [1.05]	0.82 [1.36]	407	0.91 [1.03]	1.21 [1.32]
21–24	211	0.75 [3.72]	1.16 [3.89]	88	1.80 [3.41]	2.45 [3.56]
above 24	40	0.98 [7.17]	2.14 [7.47]	13	4.28 [6.62]	5.01 [6.77]

Notes: MAE_this study_ and RMSE_this study_ denote the MAE and RMSE of RR_M_ and RR_I_, MAE_EEMD-PCA_ and RMSE_EEMD-PCA_ denote the MAE and RMSE of RR_P_ and RR_I_, and N denotes the number of windows.

**Table 5 diagnostics-13-00913-t005:** Pearson correlation coefficient (PCC) between signal quality indices and error rate of respiration rate estimated based on the EEMD-PCA method for different subjects.

Subject	R1	R2	R3	R4
Subject 01	0.14	0.70	−0.24	−0.70
Subject 02	0.30	−0.10	0.07	0.08
Subject 03	−0.22	0.07	−0.27	−0.54
Subject 04	−0.07	0.43	−0.16	−0.43
…	…	…	…	…
Subject 53	−0.32	−0.29	−0.29	−0.16
Average	0.28	0.24	0.27	0.28

Notes: R1 denotes the Pearson correlation coefficient between $Q_{R1}$ and the error rate of respiration rate estimated based on the EEMD-PCA method; R2 denotes the Pearson correlation coefficient between $Q_{R2}$ and the error rate of respiration rate estimated based on the EEMD-PCA method; R3 denotes the Pearson correlation coefficient between $Q_{R3}$ and the error rate of respiration rate estimated based on the EEMD-PCA method; R4 denotes the Pearson correlation coefficient between $\bar{S_{SQI}}$ and the error rate of respiration rate estimated based on the EEMD-PCA method.

## Data Availability

The data used in this manuscript can be downloaded from this link https://www.physionet.org/content/bidmc/1.0.0/ (accessed on 14 January 2023).

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
