# Peer review of "Whale Optimization Algorithm with a Hybrid Relation Vector Machine: A Highly Robust Respiratory Rate Prediction Model Using Photoplethysmography Signals"

_diagnostics, 2023, doi:10.3390/diagnostics13050913_

Round 1

Reviewer 1 Report

The authors propose: Whale Optimization Algorithm with a Hybrid Relation Vector Machine: A Highly Robust Respiratory Rate Prediction Model using Photoplethysmography Signals. I have some concerns and my suggestions are listed below:

1- The contribution is not adequately explained in the abstract. There is no driving force behind the essay. The information was not presented in a way that was understandable and straightforward. The main idea of the work should be emphasized in the abstract section.

2- The writers should focus on the study's main issue in the introduction and provide a Literature Review in the form of tables to identify research gaps and innovations.

3- It is crucial to enhance experimental findings, validate them, and compare them to MORE different approaches. More discussions and analyses are necessary.

4- The benefits and drawbacks of the connected works were not assessed by the authors. Please assess how their research differs from other studies in the section OF related work. What do they possess that others lack? Why or how are they superior? What is novel or new in this?

5- It is crucial to describe the WOA-HRVM’ computational complexity.

6- For experiments, nonparametric tests should be used.

7- The authors should be clear about the method's benefits and drawbacks. What are the methodology(ies) and limitation(s) used in this work? Please list benefits to daily life and go over any research limitations.

8- If you have developed any code or software, it is recommended that you provide a link to the code for other readers and to enhance the impact of the paper and its applicability

Author Response

Thank you very much for your suggestions on my article, I have followed your 8 points to improve the article, all suggestions and responses are detailed in the "attached author to respond reviewer - MDPI. docx". 

Reviewer 2 Report

In this study, authors consider both PPG signal quality and respiration signal quality to estimate respiration rate based on PPG signals and validate its accuracy and robustness on the BIDMC dataset.  Authors proposes a Whale Optimization Algorithm with a Hybrid Relation Vector Machine: A Highly Robust Respiratory Rate Prediction Model using Photoplethysmography Signals. The proposed model is interesting, and the research problem is relevant. 

In my opinion, the paper is well structured and written. The proposed paper can be accepted to the Diagnostics journal. 

Author Response

Thank you very much for your efforts and suggestions, Thank you very much.

Round 2

Reviewer 1 Report

The manuscript has been improved and the authors' comments have been substantially processed. Prior to publication, it is advised to thoroughly review the English and writing and adding your response about this issue "It is crucial to describe the WOA-HRVM’ computational complexity" in the paper contents